# Chiral DNA sequences as commutable controls for clinical genomics

Ira W. Deveson [1,2], Bindu Swapna Madala[1], James Blackburn [1,2], Chris Barker[1], Ted Wong[1],
Kirston M. Barton[1], Martin A. Smith [1,2], D. Neil Watkins[1,2,3] & Tim R. Mercer[1,2,4]

Chirality is a property describing any object that is inequivalent to its mirror image. Due to its 5′–3′ directionality, a DNA sequence is distinct from a mirrored sequence arranged in reverse nucleotide-order, and is therefore chiral. A given sequence and its opposing chiral partner sequence share many properties, such as nucleotide composition and sequence entropy. Here we demonstrate that chiral DNA sequence pairs also perform equivalently during molecular and bioinformatic techniques that underpin genetic analysis, including PCR amplification, hybridization, whole-genome, target-enriched and nanopore sequencing, sequence alignment and variant detection. Given these shared properties, synthetic DNA sequences mirroring clinically relevant or analytically challenging regions of the human genome are ideal controls for clinical genomics. The addition of synthetic chiral sequences (sequins) to patient tumor samples can prevent false-positive and false-negative mutation detection to improve diagnosis. Accordingly, we propose that sequins can fulfill the need for commutable internal controls in precision medicine.

[1] Garvan Institute of Medical Research, Sydney 2010 NSW, Australia. [2] St Vincent's Clinical School, University of New South Wales, Sydney 2010 NSW, Australia. [3] Department of Thoracic Medicine, St Vincent's Hospital, Sydney 2010 NSW, Australia. [4] Altius Institute for Biomedical Sciences, Seattle 98121 WA, USA. These authors contributed equally: Ira W. Deveson, Bindu Swapna Madala. Correspondence and requests for materials should be addressed to I.W.D. (email: i.deveson@garvan.org.au) or to T.R.M. (email: tmercer@altius.org)

Nucleic acid sequences have an inherent directionality (denoted 5′−3′) that is observed by all cellular processes, including DNA replication, transcription, and translation[1–3]. Due to this directionality, a DNA sequence is distinct from an exact copy arranged in reverse nucleotide order. This is an instance of *chirality*, a geometric property describing any object that is inequivalent to a mirror image of itself[4]. The human hand is commonly used to illustrate this concept: although the right hand is a perfect reflection of the left, the two cannot be superimposed, and therefore constitute a pair of chiral objects. Similarly, the DNA sequence 5′-ATGCATGC and the mirrored

sequence 5′-CGTACGTA are non-interchangeable, and constitute a chiral sequence pair (Fig. 1a).

For any human DNA sequence, there exists a single opposing chiral sequence. While this mirrored sequence is distinct, it shares many properties with the original human sequence, such as nucleotide composition and sequence entropy or repetitiveness (Supplementary Fig. 1a, b). Given their shared properties, chiral DNA sequences have the potential to act as proxy representations of true human sequences, and might be ideally suited for use as reference standards during genetic analysis.

**Fig. 1** Matched performance of chiral DNA sequence pairs during PCR amplification. **a** Schematic representation of a generic DNA sequence (5′-ATGCATGC) and its mirrored sequence (5′-CGTACGTA) that together form a chiral pair. **b** Schematic shows PCR primer pairs (colored arrows) targeting intervals within a synthetic human DNA sequence (*fwd*; blue) and mirrored primer-pairs targeting corresponding intervals in the chiral partner sequence (*rev*; red). In total, 14 chiral pairs of PCR amplicons were tested (Supplementary Table 1). **c** The Bar chart shows amplification efficiencies for chiral pairs of amplicons, as measured by real-time PCR (detection cycle threshold; CT). Presented values are mean ± standard deviation (n = 3). **d** Gel electrophoresis images show detection of a single *fwd* (F; blue) or *rev* (R; red) amplicon pair amplified by endpoint PCR over a gradient of magnesium concentration conditions (upper; 0–30 mM) or annealing temperatures (lower; 46–68 °C). Original gel images are provided as Source Data file

DNA reference standards are used to measure and mitigate technical biases during genetic analyses, such as clinical genome sequencing[5–7]. Existing standards can be divided into two categories, each with different advantages and limitations[7]. Reference genome materials derived from well-characterized human cell lines provide valuable process controls to evaluate analytic workflows[5,8–10]. However, human genome materials cannot be added to patient samples without causing contamination, meaning they cannot be used as internal, assay-specific controls[7]. Alternatively, artificial or non-human sequences can be used as internal spike-in controls[11–14]. However, these are necessarily distinct from human DNA sequences and, hence, do not recapitulate context- and sequence-specific variables that often confound analysis[15,16].

Chiral DNA sequences can be readily distinguished from human sequences. As a result, a synthetic chiral DNA sequence could be added to a patient DNA sample, accompany it through a diagnostic sequencing workflow, and, thereby, act as an internal control[7]. However, to constitute an ideal control, a chiral sequence must show matched performance—or be *commutable*—to its corresponding human DNA sequence during relevant bimolecular and bioinformatic processes.

Here we compare the performance of human DNA sequences to synthetic chiral partner sequences during the laboratory and bioinformatic processes that underpin modern genetic analysis. We find chiral DNA sequence pairs to be equivalent in all contexts considered, including PCR amplification, hybridization, sequencing reactions, and downstream analysis. Given their commutability to human sequences, we create synthetic chiral sequences that directly mirror a range of clinically relevant and/or analytically challenging regions of the human genome, and assess their utility as internal controls during the analysis of lung- and colorectal cancer patient DNA samples. We provide chiral DNA controls (termed *sequins*) to address the need for commutable, internal reference standards for clinical genomics[7,17–19].

## Results

**PCR amplification**. The polymerase chain reaction (PCR) amplifies DNA sequences for detection, quantification, or use in further experimental processes[20]. Given the centrality of this technique to molecular biology, we first tested whether chiral pairs of DNA sequences are amplified equivalently by PCR.

To do so, we synthesized a 2.8 kb sequence from the human reference genome (*fwd*), as well as a corresponding mirrored chiral DNA sequence (*rev*), thereby creating a chiral pair of DNA sequences (see Methods). We designed 14 pairs of primers targeting non-overlapping intervals within the *fwd* DNA template, then mirrored each *fwd* primer sequence to generate a *rev* primer pair targeting the corresponding interval in the *rev* DNA template (Fig. 1b, Supplementary Table 1). Therefore, each PCR reaction that amplified a human DNA sequence was matched by a mirrored PCR reaction amplifying the corresponding chiral sequence.

We then combined the synthetic *fwd* and *rev* DNA sequences at equal concentration in a template mixture for real-time PCR, which was performed using each pair of primers. Given the primer-pairs produce amplicons from a common DNA template, the order of amplicon detection indicates the relative amplification efficiency among *fwd/rev* PCR reactions. We found that the order of detection among *fwd* amplicons was matched by their *rev* counterpart amplicons ($\rho = 0.96$), and for each of four *fwd* sequences that failed to be sufficiently amplified for detection, the corresponding *rev* amplicon also failed (Fig. 1c). Similarly, melting temperatures recorded during this experiment were concordant between corresponding *fwd* and *rev* amplicons ($\rho = 0.78$; Supplementary Fig. 2a, b).

We also tested whether the amplification efficiency of chiral DNA sequence pairs is similarly impacted by variation in reaction conditions. We performed endpoint PCR to assess the amplification of *fwd* and *rev* targets across a gradient of annealing temperature and magnesium chloride concentration conditions (see Methods). We found that the permissible range of reaction conditions for successful amplification was matched between corresponding *fwd* and *rev* sequences (Fig. 1d), indicating that PCR amplification between paired human/chiral sequences is equivalent, and their amplification is similarly impacted by technical variables.

**Next-generation sequencing**. Next-generation sequencing (NGS) enables high-throughput determination and quantification of DNA sequences[21,22], and has become a central technique in biomedical research and clinical diagnostics[19,23]. To assess the performance of chiral DNA sequence pairs during NGS, we created eight synthetic pairs. Each pair comprised a 1.8 kb sequence retrieved from a clinically relevant position in the human reference genome (such as cancer gene exons) as well as the corresponding mirrored chiral DNA sequence. Human (*fwd*) and chiral (*rev*) DNA sequences were then mixed at equal abundance and analyzed by NGS (Illumina HiSeq), with the resulting libraries aligned to constituent *fwd* and *rev* sequences for an evaluation of sequencing performance (see Methods).

The heterogeneous sequencing coverage typically observed in NGS analysis reflects the contribution of multiple technical variables during library preparation and sequencing[15,24,25]. We found that, despite being highly variable, the distribution of sequencing coverage was similar between paired chiral sequences (Fig. 2a). For comparison, the correlation between per-base coverage profiles for paired *fwd/rev* sequences ($R^2 = 0.84$) was almost as strong as the correlation between identical *fwd/fwd* sequences analyzed in replicate experiments ($R^2 = 0.94$), while unpaired sequences exhibited no correlation ($R^2 = 0.02$; Fig. 2b, Supplementary Fig. 3a). Notably, regions that were sequenced poorly in human (*fwd*) sequences, such as sites of simple repeats or extreme GC content, were also sequenced poorly in corresponding *rev* sequences (Supplementary Fig. 3b, c), indicating that the impact of technical variables on sequencing coverage is matched between chiral DNA sequence pairs.

We next assessed the impact of sequencing errors, which are commonly introduced during sample handling, library preparation, and sequencing[16,26]. As for coverage, the distribution of different sequencing errors (nucleotide mismatches, insertions, deletions) was variable and non-random, and was closely matched between chiral DNA sequence pairs (Fig. 2c). The correlation between sequencing error frequency profiles was as strong between paired *fwd/rev* sequences (mismatch errors: $R^2 = 0.64$, indel errors: $R^2 = 0.88$) as between identical *fwd/fwd* sequences in replicate experiments ($R^2 = 0.72$, $R^2 = 0.84$, Fig. 2d, e, Supplementary Fig. 4a, b). This indicates that the sequencing errors that occur in human DNA sequences, and that confound genetic analysis, are closely recapitulated by their chiral partner sequences.

Finally, we found that other quality metrics routinely used to assess NGS, including insert size distributions, sequencing quality scores, and mapping quality scores, were also equivalent between *fwd* and *rev* sequences (Supplementary Fig. 4c–e). Together, these results demonstrate that chiral pairs of DNA sequences exhibit matched performance during library preparation and NGS analysis, with the concordance of sequencing coverage and error profiles between chiral pairs approaching that of identical human sequences analyzed in technical replicates.

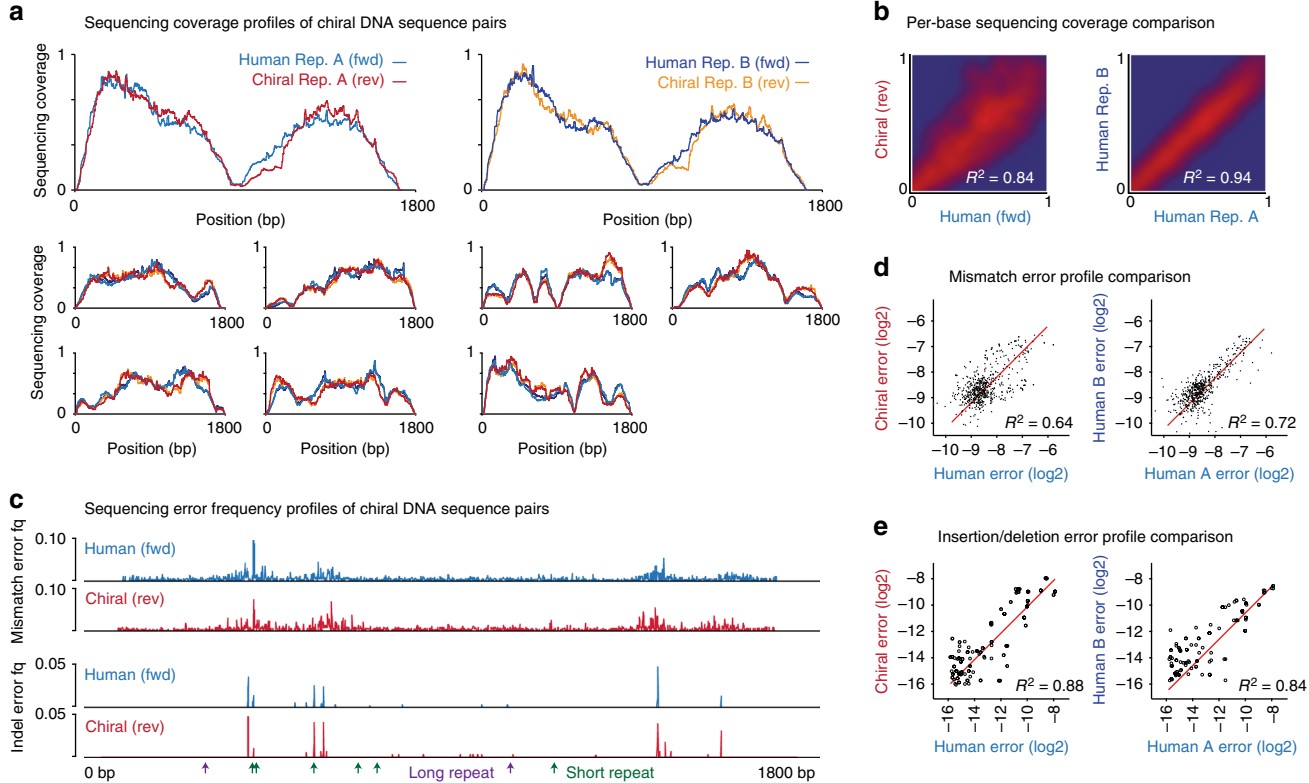

**Fig. 2** Matched performance of chiral DNA sequence pairs during next-generation sequencing. **a** Normalized sequencing coverage within synthetic chiral DNA sequence pairs ($n = 8$). Coverage profiles for human (*fwd*; blue and navy) and chiral (*rev*; red and orange) sequences are shown for two replicate NGS experiments. **b** Density scatter plots show the concordance of per-base coverage profiles between paired human and chiral (*fwd/rev*) sequences (left). For comparison, the concordance of identical human (*fwd/fwd*) sequences between replicate experiments is also shown (right). **c** Mismatch and indel sequencing error frequency profiles within one chiral pair of DNA sequences. **d, e** Scatter plots show concordance of mismatch (**d**) and indel (**e**) sequencing error frequency profiles between synthetic human and chiral (*fwd/rev*) DNA sequence pairs (left). For comparison, the concordance of identical human (*fwd/fwd*) sequences between replicate experiments is also shown (right)

**Hybridization and target-enriched sequencing.** The hybridization of DNA sequences to labeled oligonucleotide probes enables their detection, quantification, or localization within a sample[27–30]. This process is also used to enrich specific sequences during targeted NGS approaches, such as exome sequencing[31,32], wherein hybridization kinetics constitute a major source of technical bias[33]. Therefore, we next compared hybridization kinetics between chiral pairs of DNA sequences.

To do so, we synthesized a large set of 1.8 kb chiral DNA sequences ($n = 155$) that mirrored human DNA sequences. This included 68 chiral sequences mirroring "difficult" genome sites, such as GC-rich, GC-poor, or repetitive sequences. All chiral sequences were combined at equal abundance, forming a mixture of synthetic DNA sequences that collectively mirrored 279 kb of the human reference genome (see Methods). We then designed a custom panel of complementary oligonucleotide probes targeting the relevant human genome regions, as well as their chiral partner sequences. We ensured that each oligonucleotide probe targeting a human sequence was mirrored by a reverse probe targeting the corresponding chiral sequence, such that chiral sequence pairs would be captured via hybridization interactions that perfectly mirrored each other (Fig. 3a).

We then used this custom panel to perform target-enriched NGS on a combined sample containing human genomic DNA (NA12878) and the mixture of synthetic chiral DNA sequences, which differ only at sites of genetic variation in the NA12878 genome (>99% identity; see Methods). At matched sequencing depth, equivalent target coverage was achieved for human sequences and their chiral counterparts, with 93.5% and 93.7%

of captured bases reaching at least 30-fold coverage, respectively (Supplementary Fig. 5a). Sequencing coverage distributions were also correlated between paired human and chiral sequences ($R^2 = 0.86$), matching the correlation observed between identical human regions in replicate experiments ($R^2 = 0.87$), while unpaired sequences showed no correlation ($R^2 < 0$; Fig. 3b, c).

We also assessed the contribution of sequence features known to influence the hybridization of target sequences[33]. Sequencing coverage was similarly depleted at corresponding repetitive and GC-rich sequences in chiral sequence pairs (Supplementary Fig. 6a, b), while human/chiral coverage profiles remained concordant within both GC-rich ($R^2 = 0.85$) and GC-poor ($R^2 = 0.82$) regions (Supplementary Fig. 6c). Finally, we observed little correlation between coverage distributions for whole-genome and target-enriched NGS experiments performed on the same sample ($R^2 = 0.09$, Supplementary Fig. 5c, d), illustrating the dominant impact of variation introduced during hybrid enrichment that is absent during whole-genome analysis.

**Nanopore sequencing.** Nanopore sequencing measures the displacement of ionic current as DNA is passed through a transmembrane pore, enabling single-molecule, real-time sequencing of long DNA molecules[34,35]. Since this is a distinct sequencing mechanism, we next investigated whether pairs of chiral DNA sequences also perform equivalently during nanopore sequencing. We sequenced the equimolar mixture of eight synthetic chiral DNA sequence pairs (*fwd/rev*; described above) using an Oxford Nanopore MinION, and performed base-calling and read alignment to constituent *fwd* and *rev* sequences (see Methods).

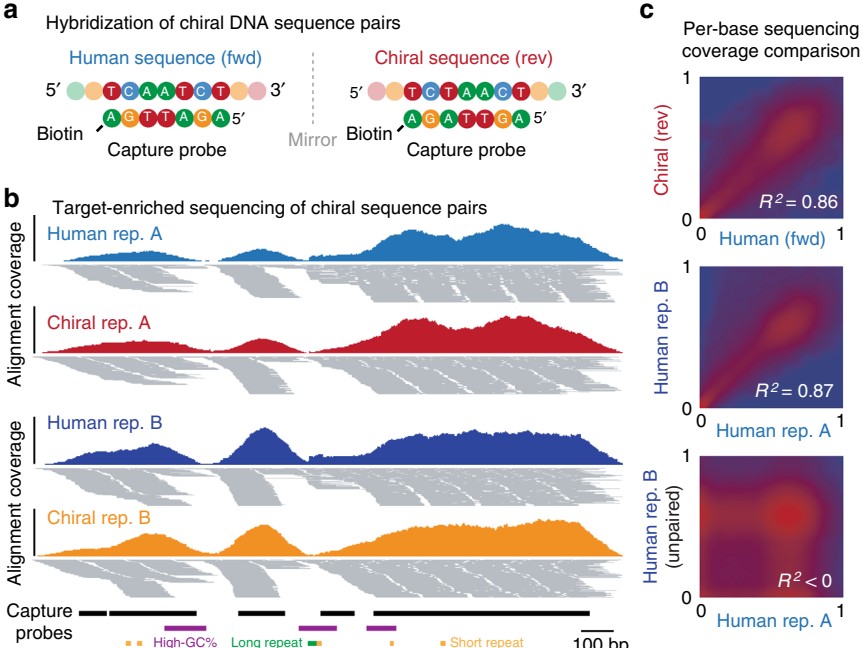

**Fig. 3** Matched performance of chiral DNA sequence pairs during target-enriched NGS. **a** Schematic shows hybridization of a human DNA sequence to a labeled oligonucleotide probe (left) and hybridization of its chiral partner sequence to a mirrored oligonucleotide probe (right). This illustrates how human/chiral sequence pairs are captured via mirrored hybridization interactions. **b** Genome browser view shows alignments from duplicate target-enriched NGS experiments analyzing a human DNA sample (blue, navy) with synthetic chiral sequences (red, orange) added. Targeted genome regions and sequence features are indicated below. **c** Density scatter plots show the concordance of per-base coverage profiles for paired human/chiral regions (upper). For comparison, the concordance of identical human/human sequences (middle) and unpaired human/human sequences (lower) between replicate experiments are also shown

We observed equivalent read-length and coverage distributions between human and chiral DNA sequences (Supplementary Fig. 7a, b). More importantly, the distribution of sequencing errors, which are common in nanopore sequencing[36], was also closely matched between chiral DNA sequence pairs (i.e., error-prone positions in *fwd* sequences were also error-prone in corresponding *rev* sequences; Fig. 4a, Supplementary Fig. 7c). This was most apparent at simple repeats, where *fwd* and *rev* sequences were similarly enriched for sequencing errors, with an equivalent dependency on repeat length (Fig. 4b). Overall, the correlation of frequency profiles for indel errors between paired human and chiral sequences (*fwd/rev*; $R^2 = 0.61$) was similar to identical human (*fwd/fwd*) sequences analyzed in technical replicates ($R^2 = 0.72$; Fig. 4c). While frequency profiles for nucleotide mismatch errors were also concordant between *fwd/rev* sequences ($R^2 = 0.42$; Supplementary Fig. 7d), the effect was weaker, suggesting the distribution of these errors during nanopore sequencing is less systematic than for indels.

**Sequence alignment**. Genome analysis typically requires the alignment of sequenced reads to the human reference genome[37,38]. Just like molecular processes, sequence alignment algorithms respect DNA directionality, and should distinguish between the members of a chiral sequence pair. Therefore, we next investigated the chiral properties of the human genome, and tested whether alignment performance is equivalent between mirrored sequences.

We first reversed each human chromosome sequence (*hg38*) to create chiral chromosome sequences that together form a mirrored reference genome (*hg38-rev*). We then simulated NGS libraries from both *hg38* and *hg38-rev*, and aligned these to a combined genome index (see Methods). From a library of 732 million paired-end reads derived from *hg38-rev*, just 133 (1.82 ×

$10^{-5}$%) were erroneously aligned to *hg38*, while the converse was true for a library derived from *hg38* ($1.05 \times 10^{-5}$% aligned to *hg38-rev*; Fig. 5a; Supplementary Table 2). The rate of cross-alignment between *hg38* and *hg38-rev* was also negligible for unpaired reads and/or shorter read lengths (Supplementary Table 3). Comparing human and chiral alignments, we found that mapping profiles were closely correlated between mirrored reference genomes ($R^2 = 0.97$; Fig. 5b, Supplementary Fig. 8a, b). Whole-genome NGS libraries from a human sample (NA12878) also exhibited negligible rates of cross-alignment to *hg38-rev* (up to 0.2%), of which the majority (>99%) of cross-aligned reads originated from low-level laboratory bacterial contamination in the samples analyzed (Supplementary Table 2). Together, this shows that almost all of the human genome is inequivalent to its mirror image (with the rare exception of very long repetitive sequences; Supplementary Fig. 8c) and that chiral DNA sequence pairs share equivalent alignability.

**Identification of genetic variation**. The identification of genetic variation is the major application of NGS in biomedical research and clinical medicine[19,23]. However, this process can be confounded by sequencing errors and coverage heterogeneity[15,16,24,26]. Given that these confounding artefacts are recapitulated between chiral DNA sequence pairs, we next tested whether synthetic chiral variants mirroring true human variants would show matched performance during variant detection by NGS.

We synthesized 87 chiral DNA sequences to represent common variants (73 SNVs and 14 indels) that are also present within the well-characterized NA12878 genome[5]. A single chiral sequence was used to represent homozygous variants ($n = 29$), while a pair of chiral sequences representing reference and variant alleles was used to emulate heterozygous genotypes ($n = 58$). We

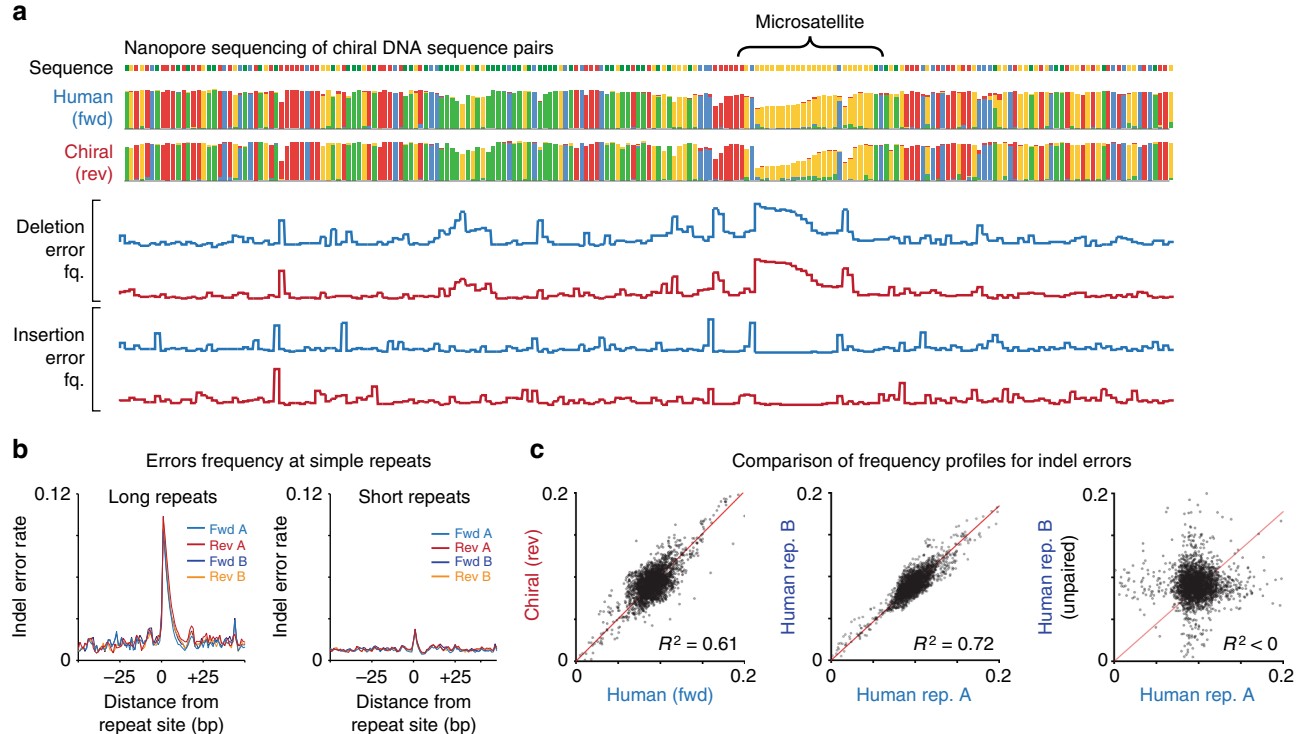

**Fig. 4** Matched performance of chiral DNA sequence pairs during nanopore sequencing. **a** Genome browser view (200 bp window) shows sequence identity and per-base insertion and deletion error frequencies within a synthetic human sequence (*fwd*; blue) and its chiral partner sequence (*rev*; red) analyzed by nanopore sequencing (Oxford Nanopore MinION). **b** Normalized indel error frequencies for human and chiral sequences aggregated w.r.t. sites of long repeats (>10nt; left) and short repeats (5–9 repeats; right). **c** Scatter plots show concordance of indel sequencing error frequency profiles between synthetic human and chiral (*fwd/rev*) DNA sequence pairs (left). For comparison, the concordance of identical human (*fwd/fwd*) sequences (middle) and unpaired sequences (right) between replicate experiments are also shown

added this mixture of synthetic chiral variant sequences to genomic DNA from NA12878 at low fractional abundance (~1%) before performing whole-genome NGS (Illumina HiSeqX; see Methods).

By aligning the output library to a combined *hg38/hg38-rev* genome index, we could distinguish reads derived from synthetic chiral sequences from reads derived from human DNA (Fig. 5c). We then reversed the sequence orientation of chiral-derived reads, while preserving Phred quality scores and paired-end relationships, so that these could be re-aligned to the *hg38* reference at the positions from which chiral sequences were originally sampled (Fig. 5c). Alignments derived from chiral sequences were then down-sampled to achieve sequencing depth equivalent to the accompanying human genome sample (Supplementary Fig. 9a), and candidate variants were identified (using GATK[39]; see Methods). This process enables a direct comparison of each synthetic chiral variant to its corresponding human variant in the accompanying NA12878 sample (Fig. 5d).

We found the sensitivity of variant detection was matched, with the same number of variants detected between human sequences and their chiral partners (85/87; *sn* = 0.98). Notably, for the two human variants that were not detected, the corresponding chiral variants were also missed (Supplementary Fig. 9b). Incremental down-sampling of sequencing libraries demonstrated that the detection sensitivity of human and chiral variants was impacted similarly by reductions in sequencing depth (Fig. 5e). Alignment qualities and variant allele frequencies (VAFs) for human and chiral variants were also concordant (Supplementary Fig. 9c, d). Finally, variant confidence (Qual) scores for human variants were correlated with Qual scores for corresponding chiral variants ($R^2 = 0.71$; Fig. 5f). Indeed, the correlation between human/chiral variants was higher than the correlation between identical human variants in replicate sequencing libraries ($R^2 = 0.61$; Fig. 5f), suggesting that the scale of technical variation between whole-genome NGS experiments exceeds the variation between paired human and chiral variants.

**Representing cancer driver mutations with chiral DNA controls.** The commutability of chiral DNA sequences to their human counterparts fulfills a key requirement of diagnostic reference standards[7]. Given they can also be readily distinguished from human sequences, synthetic chiral sequences appear ideally suited for use as internal controls in clinical genome analysis.

To demonstrate the use of chiral DNA controls, which we termed *sequins* (*seq*uencing spike-*ins*), we selected 94 recurrent, driver or clinically actionable cancer mutations for representation with synthetic chiral sequences (see Methods). To emulate the range of somatic VAFs encountered in tumor samples[40], we combined these sequins representing cancer mutations into a staggered quantitative ladder ranging from 100% to 0.1% VAF (Fig. 6a, b). We also formulated an additional "matched normal" genome mixture that included only wild-type chiral sequences against which somatic mutations can be identified. For example, the driver mutation *TP53:R273C*[41] is represented at a VAF of 6.25% in the "tumor" sequin mixture, while the *TP53* wild-type sequence alone is included in the "matched normal" sequin mixture (Fig. 6a, b).

We added sequin mixtures at low fractional abundance (~1%) to cancer genome reference samples before performing target-enriched NGS, using a custom panel that captured 134

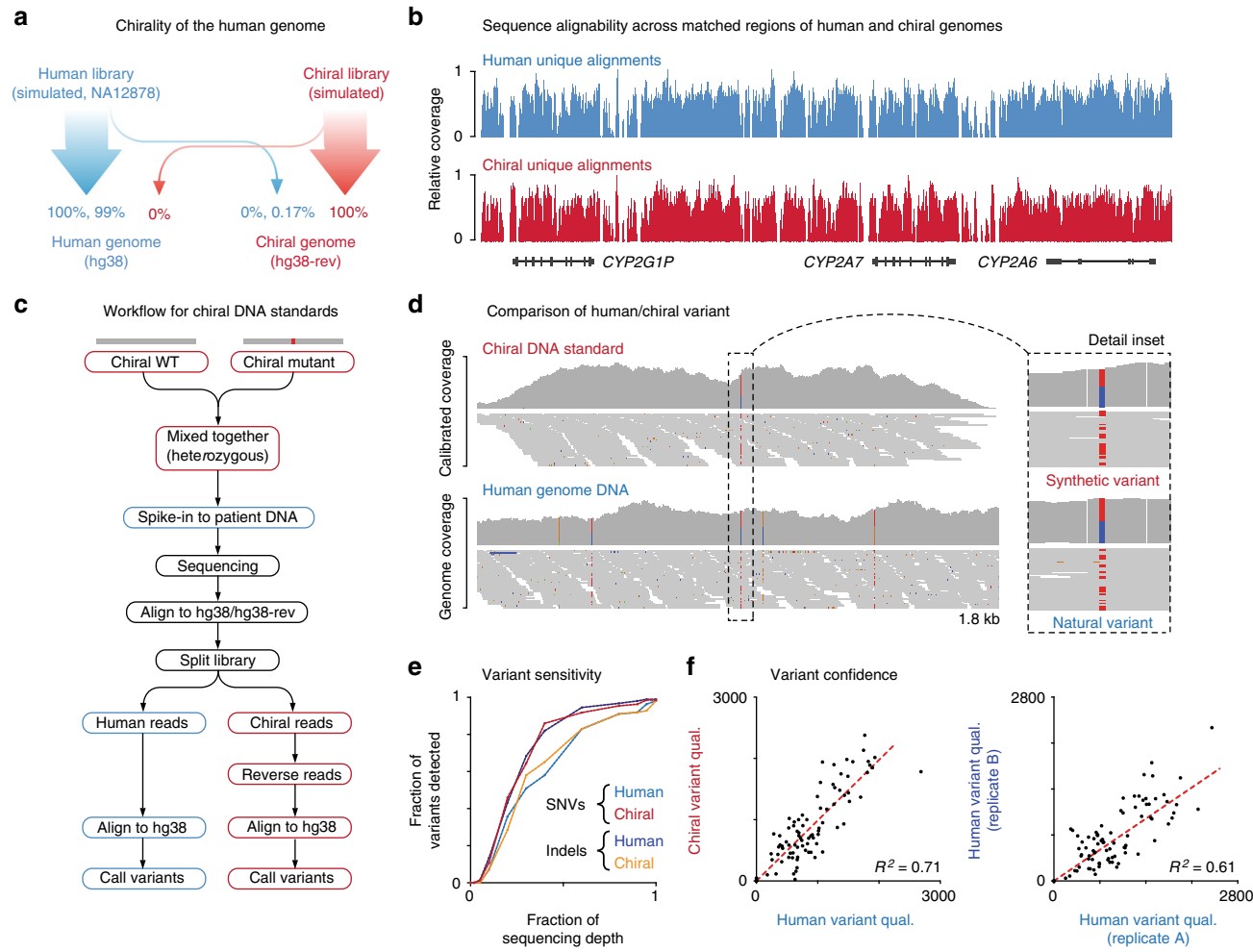

**Fig. 5** Matched performance of chiral DNA sequence pairs during variant detection. **a** Schematic summarizes the alignment rate for simulated and experimental NGS libraries to the human genome (*hg38*; blue) and a mirrored human genome sequence (*hg38-rev*; red). **b** Coverage of uniquely aligned reads (MapQ > 10) to a matched region of the human (*hg38*; blue) and chiral (*hg38-rev*; red) genome illustrates concordant mapability profile. **c** Workflow schematic describes the analysis of variants represented with chiral DNA standards, in parallel with their human variant counterparts. **d** Genome browser view showing alignment coverage from human genomic DNA (NA12878) and a matching chiral DNA standard at the site of a heterozygous single-nucleotide variant (SNV). Inset shows detail of human/chiral alignments at SNV site. **e** Plot shows variant detection sensitivity relative to sequencing depth for human/chiral SNVs (blue/red) and indels (navy/orange). **f** Scatter plots show concordance of variant confidence scores (Qual.) between human/chiral variants (left) and identical human/human variants between replicate experiments (right)

cancer-related genes, as well as their corresponding chiral sequins (see Methods). The per-base distribution of coverage among human exons and their chiral partners was strongly correlated ($R^2 = 0.95$; Supplementary Fig. 10a) and coverage of sequin variant sites was concordant with their human equivalents ($R^2 = 0.93$; Supplementary Fig. 10b). Accordingly, we were able to detect and quantify synthetic sequin variants and known variants in the human reference samples (measured independently via droplet digital PCR) with equivalent accuracy over the same quantitative range ($R^2 = 0.96$, $R^2 = 0.98$; Fig. 6c). This confirms the suitability of sequins representing cancer mutations as internal controls for tumor genome analysis.

**Analysis of patient tumor samples with chiral sequins.** To demonstrate the utility of sequin controls in a clinical context, we performed a case study analyzing retrospective tumor biopsy samples from three metastatic lung cancer patients (two lung adenocarcinomas and one small-cell lung cancer) presenting at St. Vincent's Hospital, Sydney[42] (see Methods). Sequins were added to extracted patient DNA, with the synthetic tumor and normal

mixtures added to patient tumor and matched normal samples, respectively. We analyzed these combined samples via target-enriched NGS (as above) and identified somatic variants within human exons and their chiral sequin counterparts (using Strelka2[43]; see Methods). In each patient sample, between 354 and 770 raw somatic variant candidates were detected in human exons (426 kb) and 127–148 candidates in the exonic sequin sequences (41 kb).

We analyzed these sets of variant candidates separately in the presence and absence of internal sequin controls (see Methods). In the absence of sequins, we used a best-practice variant confidence threshold to filter variants in all samples, returning 10–26 filtered variants per patient (Fig. 6d, Supplementary Fig. 10c, Supplementary Data 1). Alternatively, in the presence of sequins, we could empirically determine sample-specific thresholds for variant filtering that would retain the maximum number of true-positives, while excluding all false-positive mutations. For each sample, we determined the confidence threshold that would best distinguish true synthetic variants ($n = 61$–$68$) from erroneous variant candidates ($n = 54$–$86$) detected within chiral sequences (Fig. 6d, Supplementary Fig. 10c). This

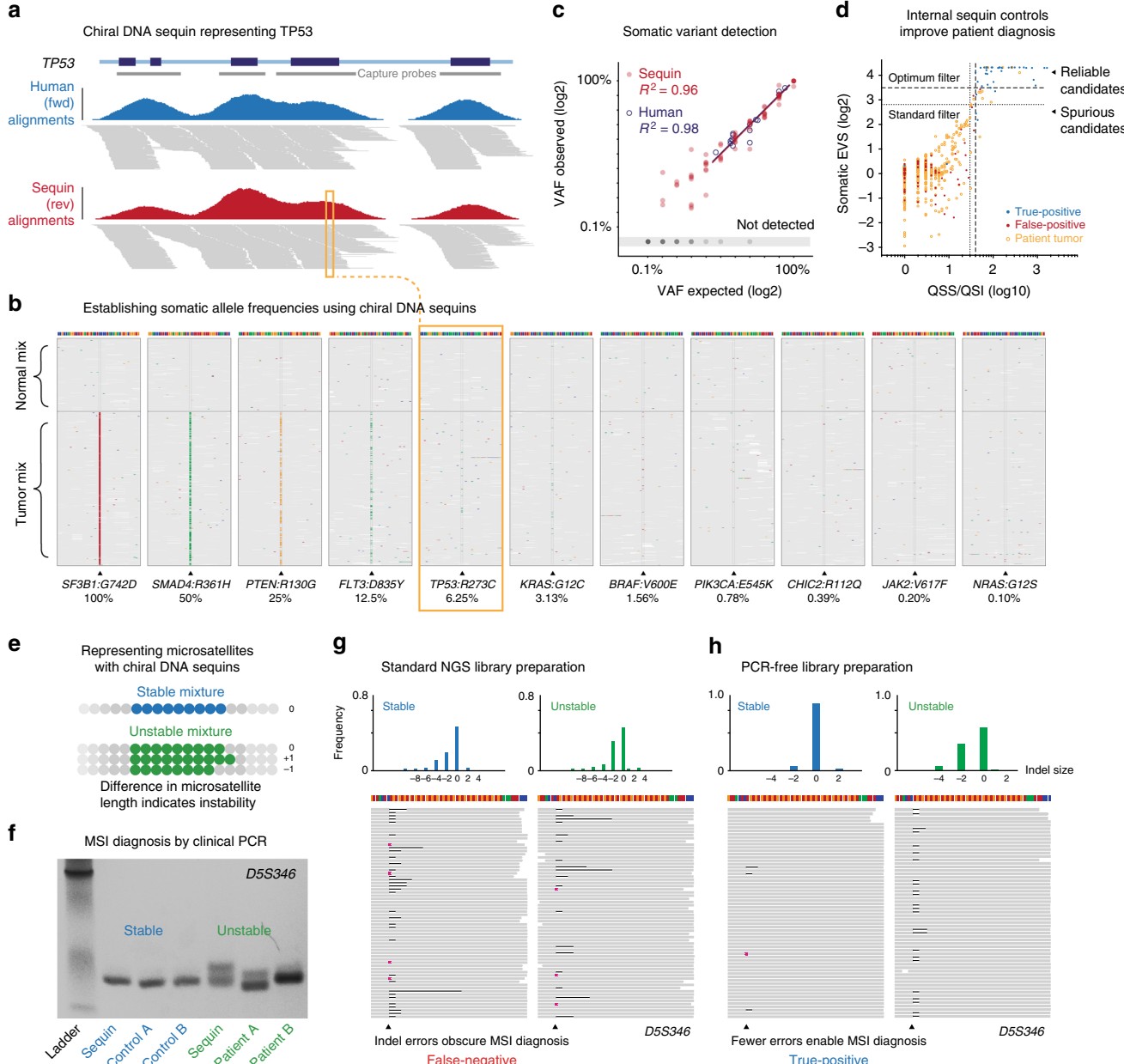

**Fig. 6** Representing cancer mutations and microsatellites with chiral DNA sequins. **a** Genome browser view showing target-enriched sequencing alignments from patient tumor DNA (blue) and chiral sequins (red) at the *TP53* locus. Gene annotation and capture probes (gray bars) are indicated above. **b** Sequencing alignments from chiral sequins at sites of cancer mutations synthetically represented across a range of variant allele frequencies (VAFs). Both normal (upper) and tumor (lower) mixtures are shown and mismatched bases colored. **c** Scatter plot shows observed vs expected VAFs for sequin variants (red) and human variants in cancer reference samples (blue). Linear regression measures detection accuracy within the VAF range 1.5–100%. **d** Scatter plot shows confidence scores (QSS/I and SomaticEVS) for somatic variant candidates in a single patient tumor sample and internal sequin controls. Optimized quality thresholds (dashed lines) exclude all false-positives (red), while retaining maximum true-positives (blue) and human variant candidates (orange). Conventional best-practice thresholds (dotted lines) are shown for comparison. **e** Schematic illustrates the design of stable (blue) and unstable (green) microsatellite sequins. **f** Size measurement of the microsatellite locus *D5S346* via PCR and gel electrophoresis for sequins (stable and unstable mixtures), MSI patient samples (*MSH6/MLH1* mutants), and matched controls. Size shifts between stable/unstable sequin microsatellites and between patient/control DNA are indicative of MSI. **g, h** Comparison of NGS alignments using standard (**g**) and PCR-free (**h**) library preparation protocols. (Lower panel) Alignments are shown for stable (left) and unstable (right) sequin mixtures at the microsatellite locus *D5S346*, with black bars and pink markers indicating the presence of deletions and insertions in sequencing reads. (Upper panel) Histograms show the relative frequency of insertions and deletions, relative to the reference microsatellite length

omitted several notable false-positive mutations, such as a missense mutation (*G796V*) in the oncogene *EGFR* that confers resistance to the targeted anti-cancer drug gefitinib[44]. By applying the optimized thresholds determined using internal sequin controls to their accompanying patient samples, we retained

2–8 confident variants per patient (Fig. 6d, Supplementary Fig. 10c, Supplementary Table 4).

To evaluate the diagnostic performance of variant detection in the presence and absence of sequins, we compared the filtered sets of patient variant candidates to calls that were generated

independently by high depth whole-genome NGS[42] (>150× coverage; see Methods). In total, 12/16 variants identified using sequins were independently validated, compared to 12/49 variants identified in the absence of internal controls (Supplementary Data 1). Of the 33 candidate variants that were excluded by reference to the sequins (but not by standard filtering), none were independently validated. Among these spurious candidates were predicted pathogenic mutations in known oncogenic and tumor-suppressor genes, such as *ALK* and *NF2*, potentially confounding patient diagnosis (Supplementary Data 2). Notably, the improvement in diagnostic specificity afforded by sequins did not reduce the sensitivity of variant detection, since no validated mutations were excluded. This clinical case study shows how internal sequin controls can be used to measure and improve diagnostic performance during patient genome analysis, highlighting their value for precision medicine initiatives.

**Representing microsatellite instability with chiral sequins.** Repetitive or low-complexity sequences are difficult to characterize and are poorly represented in genome reference materials. To demonstrate how chiral sequin controls can address this limitation, we designed synthetic sequences directly mirroring microsatellite repeats that are used in many diagnostic applications, including kinship and forensic analysis[45,46]. Increased variability at microsatellite repeats (termed microsatellite instability; MSI) is also indicative of DNA mismatch repair deficiency in multiple human cancers, informing prognosis and treatment[47–49]. However, the diagnosis of MSI using either PCR or NGS-based approaches remains challenging and no internal controls are currently available.

We therefore designed sequins representing five microsatellite loci from an established clinical reference panel used for MSI profiling[50] (see Methods). For each locus, we created a single wild-type sequin, as well as a mutated sequin differing by the insertion or deletion of a single repeat unit (Fig. 6e). The wild-type sequin alone represents a stable microsatellite, while mixed wild-type and mutant sequins emulate somatic MSI (Fig. 6e).

The current clinical standard for MSI diagnosis involves an assessment of repeat length by PCR[49]. We first compared chiral microsatellite sequins to tumor samples from nonpolyposis colorectal cancer patients with confirmed DNA mismatch repair deficiency (due to mutations in *MSH6* or *MLH1*) via PCR and gel electrophoresis (see Methods). Using clinically validated primer-pairs (Supplementary Table 4)[50], we observed equivalent amplicon size shifts between stable and unstable microsatellite sequins, as between patient tumor samples and matched controls (Fig. 6f). Therefore, MSI status, as represented by sequins, could be correctly identified using current clinical diagnostic methods.

NGS is increasingly being applied to determine MSI status in patient tumor samples[48,51]. However, the introduction of indel errors during library preparation and sequencing impedes the accurate measurement of repeat length and, thereby, MSI diagnosis. We therefore added sequins to the above patient samples and performed NGS to determine MSI status (see Methods). However, using a standard NGS library preparation procedure, abundant indel errors obscured the repeat length distinction between stable and unstable microsatellite sequins, and between patient and control samples, resulting in false-negative MSI diagnosis (Fig. 6g, Supplementary Fig. 10d).

Since PCR amplification during library preparation is a major source of error, potentially contributing to incorrect MSI diagnosis, we repeated the analysis using a PCR-free library preparation method (see Methods). This approach sufficiently reduced the frequency of indel errors to permit the resolution of repeat-length differences between stable and unstable

microsatellite sequins, and between patient and control samples, thereby enabling true-positive MSI diagnosis (Fig. 6h, Supplementary Fig. 10e). This clinical case study illustrates how internal sequin controls can be used to assess whether a given NGS test has sufficient resolution to correctly determine MSI status in an accompanying patient tumor sample.

## Discussion

Any non-palindromic DNA sequence can be reversed to form an opposing chiral sequence that is inequivalent to the original, but shares many of its intrinsic properties. Here we have shown that paired chiral sequences perform similarly in a range of biomolecular and bioinformatic processes, ranging from well-established methods, such as PCR amplification[20], to emerging technologies, such as nanopore sequencing[35].

In several instances, we found the performance of paired chiral sequences to be more similar than the performance of identical sequences between replicate sequencing assays. This affirms the equivalence of chiral DNA sequence pairs, but also serves to illustrate the impact of technical variation between repeated experiments. The scale of technical variation is greater between alternative technologies, protocols, or laboratories, confounding genetic analysis and contributing to the risk of misdiagnosis[16,25,33].

Reference standards can measure and mitigate the impact of technical variation to improve genetic diagnosis[7]. However, a major limitation is the necessary compromise between commutability (i.e., similarity of standard to sample) and the requirement for standards to be readily distinguished from patient DNA to enable their use as internal controls. Indeed, the majority of existing reference materials are extensively characterized human DNA samples that are limited to use as external process controls[5,7,9,10].

We propose that the invention of chiral DNA controls, termed *sequins*, resolves this compromise, since sequins are commutable to corresponding sequences in the human genome, yet can be easily distinguished from human DNA during sequencing analysis. We have shown that sequins can be added to patient DNA samples, providing internal controls that directly measure errors and biases that accrue during sample handling, assay execution, and bioinformatic analysis[15,16,25,26,33,52–54]. Importantly, we found that patterns of coverage heterogeneity and sequencing errors—the chief causes of false-negative and false-positive results during variant discovery, respectively—were recapitulated between human sequences and their corresponding chiral sequins. These artefacts are complex and overlapping, and vary across different regions of the human genome, emphasizing the value of sequence-matched controls that can measure the impact of technical variation on a given sequence, in a given assay. Sequins can be used to rapidly assess operational performance and gauge the quality of accompanying samples. Low-quality samples, such as clinical specimens that are excessively degraded by formalin fixation[55], can be easily identified by comparison to internal sequin controls, and technical artefacts can be disentangled from meaningful signal, such as mutation signatures in human tumors[56].

The chiral design principle is also simple and flexible. Synthetic chiral sequences can be created to represent almost any human DNA sequence, including analytically challenging loci (such as microsatellite repeats) that are otherwise difficult to reliably assess[48,51]. By preserving sequence context and nucleotide composition, sequins provide faithful analytic proxies for these features. Moreover, the catalog of matching chiral standards can be easily expanded as the inventory of clinically informative human genome regions continues to grow.

The benefits of using sequins, or any other spike-in control, during NGS experiments must be weighed against the cost of sequencing reads that are necessarily sacrificed for their analysis. This limitation is of minimal importance during whole-genome NGS, where no more than ~1% of reads must be sacrificed in order to achieve matched stoichiometry between chiral standards and the diploid human genome. During targeted sequencing approaches, a larger fraction of the sequenced library may be sacrificed, with this depending on the collective size of captured genome regions relative to captured controls, and therefore differing for alternative capture panel designs. For a typical design, users can attain the benefits to performance and reproducibility described above at a <5% increase in sequencing cost.

The diagnosis of human mutations by genome (and exome) sequencing suffers from a high false-positive rate and an undetermined false-negative rate, with the resulting impact of misdiagnosis during clinical care unknown[16,57]. Accordingly, the development of DNA reference standards that can evaluate diagnostic performance is considered a prerequisite for the maturation of genome sequencing into routine clinical practice[17–19]. By providing commutable internal controls, sequins promise to improve the accuracy and robustness of clinical genome sequencing and, thereby, help facilitate the realization of precision medicine. We offer sequins as a validated resource for the genomics community. For further information or to request an aliquot please visit www.sequinstandards.com.

## Methods

**Design of synthetic chiral DNA sequences.** To assess the performance of chiral DNA sequence pairs, we created pairs of synthetic sequences that perfectly mirrored each other. To ensure chiral pairs had the attributes of real human sequences, we retrieved continuous subsequences from the human reference genome (hg38). These were not edited or shuffled, and represent perfect synthetic copies of real human sequences (fwd), while their chiral partner sequences (rev) were created simply by arranging fwd molecules in reverse nucleotide order.

After confirming the similar performance of matched fwd/rev synthetic DNA sequences in various assays (see below), we expanded our scope, creating a large catalog of 1.8 kb synthetic rev molecules, each the chiral equivalent to a subsequence within hg38. Rather than generating synthetic fwd sequences, these were compared to real human DNA samples, with rev sequences only differing from their corresponding genome regions at the positions of genetic variants in a given human sample. Sequences selected for synthesis were centered on the following genomic features: sites of common genetic variants (NA12878), exons of cancer-associated genes (COSMIC), local regions of low (<20%) or high (>70%) GC-content, and clinical microsatellite marker sequences (Bethesda panel)[50]. For this study, we created and surveyed a total of 223 × 1.8 kb synthetic chiral sequences, covering 450 kb of genome sequence and encompassing 252 variants/mutations. Synthetic sequences and related information can be found at www.sequinstandards.com.

**Synthesis and handling of synthetic chiral DNA sequences.** We commissioned the synthesis and Sanger-sequencing validation of all synthetic molecules by a commercial vendor (ThermoFisher-GeneArt). Synthetic chiral sequences were amplified by bacterial culture, excised by restriction enzyme digest, quantified by UV fluorometry (Thermofisher Qbuit), and combined into a variety of larger mixtures using a liquid-handling robot (Eppendorf). For experiments involving synthetic fwd/rev pairs, all sequences were combined at equal concentration. In the larger mixtures comprising rev molecules representing genetic variants and disease mutations, different genotypes were emulated by combining synthetic molecules representing reference and variant alleles in precise ratios (as described previously[14]). Heterozygous variants were represented by paired reference/variant alleles at equal abundance and homozygous variants were represented by the variant allele alone. To emulate the heterogeneous somatic VAFs encountered in tumor samples, we created a ladder ranging from VAF = 100% to VAF = 0.1%, serially diluting variant alleles such that each step on the ladder was at half the frequency of the previous step, with 7–9 somatic variants at each level. In addition, we created a separate mixture in which only the reference sequence for each site was included. This mixture provides a wild-type background against which somatic mutations (encoded in the first mixture) can be called, emulating to the popular approach of matched tumor/normal sample analysis.

**PCR amplification of DNA chiral pairs.** To test the amplification efficiency of chiral pairs of DNA sequences, we designed primers targeting mirrored PCR amplicons within a single 2.8 kb fwd/rev pair of sequences (see above). Specifically, we designated 14 × PCR amplicons (120 bp each) within each template, placing these such that every amplicon in the fwd template had an exact rev equivalent, bookended by primers (20 bp each) that perfectly mirrored each other (Supplementary Table 1). We combined the synthetic fwd and rev DNA templates at equal concentration (~5.0 × 10⁻⁴ ng/μL) and performed real-time PCR with each primer pair. Real-time PCR was performed using Power SYBR Green PCR Master Mix (Life Technologies) and 2 μM primers (Integrated DNA Technologies) in triplicate on an Applied Biosystems 7900HT Fast Real-Time PCR System, under standard conditions. Cycle threshold (CT) and melting temperate values were recorded for each reaction. Because they were amplified from a common template, the order of detection among fwd/rev amplicons provides a direct measure of the relative efficiencies within each orientation category.

We then performed endpoint PCR (40 cycles) to detect paired chiral amplicons under a range of reaction conditions. We separately manipulated the annealing temperature across a 30–66 °C range and magnesium chloride concentration across a 0–30 mM range (at 60 °C). PCR was performed using KAPA HiFi Hotstart Ready mix (temperature experiment) or Taq Flexi DNA Polymerase (magnesium experiment) on a BioRad thermo-cycler, and PCR products were visualized by gel electrophoresis. In this way, we determined the minimum and maximum thresholds for temperature and magnesium concentration, outside of which a given amplicon could not be amplified sufficiently for detection.

**Performance of DNA chiral pairs during conventional NGS.** To compare the performance of chiral pairs of DNA sequences during NGS analysis, we combined 8 × 1.8 kb fwd/rev pairs (see above) at equal abundance and analyzed this mixture via a conventional short-read sequencing workflow. Duplicate libraries were prepared using a Nextera XT Sample Prep Kit (Illumina) according to the manufacturer's instructions. Prepared libraries were quantified on a Qubit system (Invitrogen) and verified on an Agilent 2100 Bioanalyzer. Paired-end sequencing was performed on an Illumina HiSeq 2500 machine housed at the Kinghorn Centre for Clinical Genomics (https://www.garvan.org.au/research/kinghorn-centre-for-clinical-genomics). The resulting libraries were trimmed using TrimGalore (v0.4.1; https://github.com/FelixKrueger/TrimGalore) then aligned to a reference index of all constituent fwd/rev sequences using BWA-mem (v0.7.16)[37] at default parameters. Coverage and sequencing error profiles were derived using bamtools piledriver (v2.2.2; https://github.com/pezmaster31/bamtools) and compared either on a per-base, local sliding window (40 bp) or whole-sequence basis, or with respect to aggregated features (e.g., simple repeat sites). Other quality control metrics were retrieved using the Picard toolkit (v2.14; https://broadinstitute.github.io/picard/).

**Performance of DNA chiral pairs during target-enriched NGS.** To perform target-enriched NGS analysis, we first commissioned the manufacture of a custom gene panel (Roche-NimbleGen), targeting genome regions represented by synthetic chiral standards, as well as 134 cancer-related genes (encompassing all cancer mutations that were synthetically represented by chiral standards). For chiral sequences anchored to common germline variants or challenging genome sites (see above), the whole 1.8 kb genome region was included in the capture design, whereas for cancer genes, only exonic sequences were captured (analogous to the design of an exome sequencing panel). From this design, we retrieved the sequences for all oligonucleotide probes targeting a region represented by synthetic chiral standards. For each, we created an exact reverse-orientation copy and included this in our custom design at equivalent frequency to its fwd-orientation counterpart. In this way, human sequences and their chiral pairs were captured via hybridization interactions that perfectly mirrored each other.

We performed targeted sequencing on human DNA samples combined with synthetic chiral sequences, according to an established protocol (Roche Double Capture Technical Note, August 2012). Briefly, samples were fragmented using NEBNext dsDNA Fragmentase (M0348) and quenched with 0.5 M EDTA. Fragmented samples were purified with 1.8× pre-warmed Agencourt AMPure XP beads, eluted and used as input for NGS. Libraries were prepared with a KAPA Library Preparation Kit (Illumina platform KR0935—v2.14), in conjunction with SeqCap Adapter Kits (Roche-NimbleGen), as per the manufacturer's protocol, with 10 cycles of PCR amplification. Purified libraries were quantified on an Agilent 2100 Bioanalyzer, before performing paired-end sequencing, as above. Reads were trimmed, aligned, and coverage and error profiles were derived and compared, as above.

**Performance of DNA chiral pairs during nanopore sequencing.** To compare the performance of chiral pairs of DNA sequences during nanopore sequencing, the mix of 8 × 1.8 kb synthetic fwd/rev pairs (see above) was analyzed on an Oxford Nanopore MinION instrument. Libraries were prepared with a LSK108 kit (1D ligation) according to the manufacturer's instructions. Duplicate experiments were performed on separate flow cells (R9.5). Libraries were base-called using ONT Albacore Sequencing Pipeline Software (version 1.2.6). The resulting base-called reads were aligned to constituent fwd/rev sequences using minimap2 (v2.7)[58], and coverage and error profiles were derived and compared, as above.

**Genome chirality**. To assess the directional specificity of human DNA sequences during sequence alignment, we first created a reverse-orientation copy of the human reference genome (*hg38-rev*), in which each chromosome and scaffold from *hg38* is arranged backwards. We used ART (v2.5.8)[59] to generate simulated NGS libraries from *hg38-rev*, as well as *hg38*, and aligned these to a combined genome index using BWA-mem (v0.7.16)[37] at default parameters (Supplementary Table 2). We then varied the parameters of this simulated experiment, manipulating read-length and read-pairing status (Supplementary Table 3). Next we tested the alignment of two experimental WGS libraries from human samples (NA12878), obtained from the Kinghorn Centre for Clinical Genomics (https://www.garvan.org.au/research/kinghorn-centre-for-clinical-genomics), to the *hg38/hg38-rev* genome index (Supplementary Table 2).

**Detection and analysis of germline mutations**. A mixture of synthetic chiral standards emulating germline variants (see above) was added to fresh genomic DNA from NA12878 at low fractional abundance (~1%) before performing library preparation (Illumina TruSeq Nano) and paired-end WGS on an Illumina HiSeqX machine (>30× coverage). Library preparation and sequencing was carried out in duplicate by the clinically accredited Genome.One sequencing facility (https://www.genome.one/).

The resulting libraries were trimmed using TrimGalore (v0.4.1; https://github.com/FelixKrueger/TrimGalore) and aligned to a combined *hg38/hg38-rev* genome index (see above) using BWA-mem (v0.7.16)[37] at default parameters. In this scenario, reads from human DNA align to *hg38* and reads from chiral sequences to *hg38-rev*, allowing them to be easily partitioned. We then used a purpose-built software package (Anaquin[60]) to reverse the sequence orientation of all chiral-derived reads, while preserving Phred quality scores and read-pair relationships. After their reversal, chiral-derived reads were re-aligned as above, but now align to *hg38* at the same position of their corresponding human sequences, enabling a direct comparison in the same genomic context.

Chiral alignments were down-sampled to achieve equivalent sequencing coverage to the accompanying human genome, and then human and chiral alignments were analyzed in parallel with a generic variant calling pipeline. Specifically, PCR duplicates were removed using Picard (v2.14), indel realignment was performed using GATK[39] (v3.8), and variants were identified using GATK HaploytypeCaller (v4.0). To assess the impact of depreciating sequencing coverage on variant detection, libraries were incrementally down-sampled before repeating variant calling as just described. Attributes of paired human/chiral variants were obtained using GATK VariantsToTable (v4.0).

**Detection and analysis of somatic mutations**. To validate our mixture of synthetic chiral standards emulating cancer driver mutations (see above), the mixture was added to fresh DNA from cell-line based cancer genome reference materials (Horizon-Discovery: EGFR Multiplex standard & Structural Multiplex standard) at low fractional abundance (~0.5%). These combined samples were analyzed by target-enriched sequencing, as above. The resulting sequencing libraries were trimmed, aligned, partitioned, and processed, as above. We recorded the coverage and VAF for each synthetic variant. To evaluate the accuracy of variant quantification, we compared observed VAFs to expected VAFs from our mixture design by linear regression. We also performed the same analysis for known human mutations harbored by the accompanying cancer genome reference materials, which have been independently quantified by droplet digital PCR (https://www.horizondiscovery.com/reference-standards/q-seq-hdx), finding equivalent quantitative accuracy for synthetic and human mutations over the same frequency range.

Next we assessed the utility of chiral standards representing cancer driver mutations for the analysis of patient tumor samples. We analyzed retrospective tumor biopsy samples (tumor cellularity ≥ 20%) from metastatic lung cancer patients (two lung adenocarcinoma and one small-cell lung cancer) that were collected for a separate study[42], with approval of ethics committees at participating institutions (St Vincent's Hospital and Garvan Institute). Samples were processed for DNA extraction from the frozen cell suspension using the Qiagen (Hilden, Germany) DNEasy kit as per the manufacturer's instructions, and then checked for quality, purity, and integrity before adding chiral DNA standards. Germline DNA was obtained using the same methodology from peripheral blood mononuclear cells. Chiral standards were added to patient DNA samples at low fractional abundance (~0.5%), with the synthetic tumor and normal mixtures added to patient tumor and matched normal samples, respectively. The combined samples were analyzed by target-enriched sequencing, as above.

Human and chiral alignments were trimmed, aligned, partitioned, and processed, as described above, and then analyzed in parallel with a best-practice pipeline for somatic mutation discovery. Specifically, PCR duplicates were removed using Picard (v2.14), indel realignment was performed using GATK[39] (v3.8), and potential somatic mutations were identified using Strelka2 (ref. [43]) (v2.8.4), and then filtered according to Strelka2 default confidence thresholds. Because the identity of every base within synthetic chiral standards is known, erroneous variant calls (FPs) can be distinguished from true variants (TPs). This information allowed us to optimize filtering thresholds (rather than using defaults) for each experiment, selecting minimum values for the scoring criteria provided by Strelka2 (QSS/QSI and SomaticEVS) that would exclude all FPs, while maximizing the number of TPs and human variant candidates that were retained (this process is visualized in

Fig. 6d and Supplementary Fig. 10c). These optimum filtering thresholds were determined and applied separately for each individual sample. After filtering, either by the standard or optimized approach, variant candidates were queried against raw somatic variant calls generated independently by deep (>150×) whole-genome NGS performed on the same sample pairs[42], for orthogonal validation. Variants of interest were evaluated for pathogenicity using the Ensembl Variant Effect Predictor tool, via the web interface (https://asia.ensembl.org/Tools/VEP).

**Analysis of MSI**. To perform PCR profiling on the Bethesda clinical microsatellite markers, and their corresponding chiral standards, we followed established guidelines[50]. For each human microsatellite, we used standard clinical primer-pairs and designed mirrored primer-pairs to target the corresponding chiral microsatellite (see Supplementary Table 4). We performed endpoint PCR (40 cycles) with each primer pair, profiling stable and unstable microsatellite mixtures separately, as well as MSI reference samples from NIBSC (MLH1/MSH2 Exon Copy Number Reference Panel. NIBSC code: 11/218-XXX) and healthy human controls. PCR was performed using KAPA HiFi Hotstart Ready mix on a BioRad thermocycler and PCR products were visualized on an acrylamide gel. Size discrepancies between matched samples, for a given microsatellite marker, are indicative of MSI.

We also analyzed chiral microsatellite mixtures by conventional NGS (Nextera XT Sample Prep kit), and as a combined sample with human DNA (NA12878), via a PCR-free library preparation (KAPA HyperPlus PCR-free kit). Reads were trimmed, aligned, partitioned, and processed as above. To assess microsatellite length, we retrieved every individual read that spanned a give repeat site, with a minimum 4 bp overhang at either end. From each read we recorded the length of the contained repeat sequence, with these together forming a distribution around the true length of the microsatellite sequence due to confounding indel errors. Discrepancies in size distributions between matched samples for a given microsatellite marker are indicative of MSI.

**Statistical analysis and graph plotting**. GraphPad Prism (GraphPad Software, La Jolla, California, USA) (v7) and R (v3) were used to generate plots and perform statistical calculations presented in figures and main text.

**Reporting Summary**. Further information on experimental design is available in the Nature Research Reporting Summary linked to this article.

## Data availability

All DNA sequencing libraries used in this study have been deposited to the NCBI Sequence Read Archive (SRA), under the Bioproject: PRJNA520809. Associated data files, including synthetic sequences and variant annotations, are available by request via www.sequinstandards.com. Sequins are freely available for not-for-profit academic research.

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

## Acknowledgements

We thank our colleagues M. Cowley, M. Pinese, and S. Hardwick for useful discussions during experimental design and manuscript preparation. We acknowledge the following funding sources: NHMRC grants APP1108254 and APP1114016, and Cancer Institute NSW Early Career Fellowship 2018/ECF013 (to I.W.D.). The contents of the published materials are solely the responsibility of the administering institution, a participating institution or individual authors, and they do not reflect the views of the NHMRC or CINSW.

## Author contributions

I.W.D. and T.R.M. conceived the project, designed chiral sequences, and devised the experiments. B.S.M., C.B. and J.B. processed synthetic constructs and assembled mixtures. B.S.M. and J.B. performed library preparations and capture enrichment for NGS experiments, and performed chiral PCR experiment. C.B. performed microsatellite PCR experiment. K.M.B and M.A.S. performed nanopore sequencing. D.N.W. provided DNA samples from lung cancer patients. I.W.D. and T.W. performed data analysis. I.W.D. and T.R.M. prepared the manuscript, with support from all coauthors.

## Additional information

**Competing interests:** The Garvan Institute of Medical Research has filed patent applications on techniques described in this study. The authors declare no competing interests.

