## [Peer Review File · Nature Communications]

Reviewers' comments:

Reviewer #1 (Remarks to the Author):

Review of "Chiral DNA sequences as commutable reference standards for clinical genomics"

Overview:

This paper describes the principles for developing molecules to be used as experimental controls in genomics experiments. The essential principle presented is to use "reversed," chiral, mirror sequences of genomic DNA as commutable experimental controls; that is, controls that match the characteristics and exhibit the same performance properties as the endogenous molecules being measured or studied when assayed with like assays. The principle of such chiral molecules as experimental controls has been described before by some of these authors, and this work extends the concept presentation with a thorough study to establish the commutability with a wide variety of sequencing assays and PCR amplification. The sequencing assays include the bioinformatic, or "Dry Lab" analytical process, a critical consideration.

The methods used to establish the commutability of the controls are well designed and innovative. Overall, the work is well thought-out, carefully done, well-presented, and thorough. This work will be of interest to the readers of Nature Communications.

The authors have presented excellent scientific work to evaluate the functionality and commutability of the controls for difficult genomic regions and contexts; they report compelling results on the microsatellite instability analysis using the PCR amplicon shift method, and then address the inability (negative result) to reliably detect the MSI with standard library-prep NGS by using a PCR-free method, showing promise for the utility of the controls in a difficult-to-sequence genomic context. This represents an advance over previous published work.

Significant considerations:

Strong Policy-level Concerns

This is a really strong paper -- It's a fine work with little to quibble with methodologically or pedagogically. The evaluation is thorough and the results are compelling.

My two concerns are around policy (for lack of a better term).

I am concerned with the (in my opinion) casual use of the term "Reference Standard." We can and must aspire to more rigor when talking about the basis for validation and developing rigorous evidence in bioscience. Using the term "Standard" in this context creates the impression that the community's needs are met -- and they are not fully satisfied unless the molecules are made available to all, for all applications.

This paper presents strong, innovative development and research results in the context of academic application. However, the authors don't present a clear path to unrestricted availability of the materials and methods for all other researchers to readily make accurate use of -- the community can't realize the benefits of standards unless the principles of broad and unrestricted availability, authoritative sourcing, and community acceptance are in place.

More on the word "Standard"

The authors use the term "standard" in the title and throughout the manuscript, without presenting a context frame for that heavily overloaded word. I suggest that the authors use the term "controls" or

"control molecules" and that they avoid the often ambiguous term "standard."

If they believe using the term "standard" is appropriate, they should frame it explicitly, defining what they mean by that term:

- The term "reference standard" is used in other laboratory medicine contexts; I suggest review of this database entry from the CLSI:

<http://htd.clsi.org/listterms.asp?searchdterm=standard&button=Submit>

- see, for instance this note under the definition of the word "standard:"

§ reference standard – a standard, generally having the highest metrological quality available at a given location or in a given organization from which measurements made there are derived (VIM93)

§ This definition is so ambiguous as to be nearly meaningless (what is "highest metrological quality"?), and again, suggest that the authors be explicit in the presentation of their fine work in developing control molecules that are both orthogonal to the molecules under study and established as commutable.

More on Control Availability

- The paper presents these controls as a scientifically interesting resource for the readers. I note that the website sequin.xyz, presented as a data resource by the authors in this paper, offers aliquots of the controls for use in non-profit research. I suggest that the authors make the availability clear to the reader in the body of the paper.

- I offer my strongest recommendation that the authors make the controls universally available, without restriction. Restricted use is inconsistent with principles of transparent and reproducible research, and antithetical with the concept of a "standard."

Suggestions for additional references to relevant work:

The below noted papers describe prior work in using "spike-in" controls for performance assessment of NGS variant detection -- these would be useful additional references:

Sims, D. J., Harrington, R. D., Polley, E. C., Forbes, T. D., Mehaffey, M. G., McGregor, P. M., ... Lih, C.-J. (2016). Plasmid-Based Materials as Multiplex Quality Controls and Calibrators for Clinical Next-Generation Sequencing Assays. *The Journal of Molecular Diagnostics*, 18(3), 336–349.
<https://doi.org/10.1016/j.jmoldx.2015.11.008>

Lincoln, S. E., Zook, J. M., Chowdhury, S., Mahamdallie, S., Fellowes, A., Klee, E. W., ... Lincoln, S. E. (2017). An interlaboratory study of complex variant detection. *BioRxiv*, 301–312.
<https://doi.org/https://doi.org/10.1101/218529>

Minor corrections:

- Page 3: "reaction that amplifying"
- Page 6: "depreciating" -- perhaps "deprecating" or "downsampling"
- Page 7: "microsatellites loci" -- should be "microsatellite loci"
- Page 8: "technically identical" -- suggest "repeated"
- Page 8: "measure and mitigate" -- suggest "assess and mitigate"
- Page 8: "chiral sequences are both commutable..." -- perhaps "chiral sequences are demonstrated here to be commutable..."
- Page 10: State method for concentration determination when pooling controls (UV spectrophotometry?)
- Page 13: "MSI reference samples from NSBCI..." should be "MSI reference samples from NIBSC..."

Reviewer #2 (Remarks to the Author):

The manuscript "Chiral DNA sequences as commutable reference standards for clinical genomics" by Deveson et al. is well structured and written. The authors introduce a very interesting approach using chiral DNA as a standard for next generation sequencing with an emphasis on clinical sequencing. They nicely show that chiral DNA closely shares all tested properties with the corresponding "mirrored original". Nevertheless, when looking closely, one can see some differences between the performance of the original sequence and its chiral twin. These differences are throughout the study often not adequately mentioned and rarely discussed. One of the biggest challenges for clinical routine sequencing is the limited availability of fresh frozen material. Often, samples are directly fixed in formalin and embedded in paraffin (FFPE) for routine assays. This procedure leads to ample artifacts in NGS sequencing. It would be nice to read what the authors think of this and if there is a possibility to mimic this effect with chiral DNA as well. Apart from this there are multiple points that should be addressed to strengthen the story:

1. It has been shown that mutational signatures as well as sequencing errors are context dependent (Alexandrov et al. Cell Rep 2013; Allhoff et al. BMC Bioinformatics 2013), in chiral DNA this context is reverted, I would expect this to be reflected on some properties and would like to see it, at least, discussed in this manuscript.
2. I could not find the used sequences on <https://sequin.xyz> (only RNA files in the download section, this manuscript is exclusively dealing with DNA).
3. R2-Differences in Figure 2 are not negligible. The general coverage profiles seem to be comparable but local differences remain. Do these have distinct properties?
4. Did the library of synthetic regions (for hybridization tests) contain SNP-Positions? Did they also synthesize the minor alleles?
5. Did they try aligning the chiral counterpart of the exact same library rather than simulating two independent ones? Does the R2 increase?
6. Where do the NGS-reads from the human sample, that map wrongly to the chiral genome actually map on the reference genome when aligned against the forward genome only? Are there patterns? 0.167% is not negligible especially if this is non-random.
7. The variants in Figure 6b are barely visible. It might be better to show stacked bar plots to show the proportion.
8. The idea of a dilution row for different VAFs is very good but unfortunately the sequencing context is not fixed since different VAFs are represented in different genes (at least in Figure 6).
9. It would be nice to see how the chiral standard performs at sides of Indel's and especially CNVs for cancer genomes, especially deletions that are very hard to determine in target enriched approaches.
10. It would be nice to get an overview of the actual read counts of all variants (TP as well as FP for both sample and chiral) from the real cancer samples?

Reviewer #3 (Remarks to the Author):

The authors have done a very comprehensive job at evaluating the use of chiral molecules as DNA standards for clinical applications. While these results are not surprising, the need to validate standards is quite clear. I generally have no concerns with the work being done. However I would like the authors to spend a bit more time discussing a few points:

1- Chiral DNA as opposed to scrambled DNA or synthetic DNA: The authors indicate that a chiral DNA molecule retains critical context information. This is a reasonable statement but it raises the question of how important context actually is. Certainly, composition has an effect on sequencing performance (GC content for example) but composition can be recapitulated via synthetic or scrambled sequences. Is there any evidence that synthetic or scrambled sequencing have notable different performance? This is a fairly minor question but worth including a few sentences to address.

2 - Cost. This is the more important of the questions the authors should address. The authors show improved specificity for true variants at different frequencies. This is a good thing. But it seems that including controls for each and every target (as seems to be what is suggested for capture experiments) would take a large portion of the sequenced bases. This will increase the cost of sequencing to a particular target coverage. Variants are currently validated via low cost Sanger methods. How different are the costs of large scale validations vs taking up sequencing real estate with control sequences.

3- I see little use in the section on nanopore. The main thrust of the use of these controls seems to be for clinical applications. Nanopore must make massive changes before this technology could be used for most clinical applications.

REVIEWER 1

1.1 I am concerned with the (in my opinion) casual use of the term "Reference Standard." We can and must aspire to more rigor when talking about the basis for validation and developing rigorous evidence in bioscience. Using the term "Standard" in this context creates the impression that the community's needs are met -- and they are not fully satisfied unless the molecules are made available to all, for all applications.

This paper presents strong, innovative development and research results in the context of academic application. However, the authors don't present a clear path to unrestricted availability of the materials and methods for all other researchers to readily make accurate use of -- the community can't realize the benefits of standards unless the principles of broad and unrestricted availability, authoritative sourcing, and community acceptance are in place.

The authors use the term "standard" in the title and throughout the manuscript, without presenting a context frame for that heavily overloaded word. I suggest that the authors use the term "controls" or "control molecules" and that they avoid the often ambiguous term "standard."

If they believe using the term "standard" is appropriate, they should frame it explicitly, defining what they mean by that term.

We appreciate the reviewer's concerns about our use of the term "standard" and have therefore substituted this term with "controls" where possible. Notably, we have modified the title of our article to:

Chiral DNA sequences as commutable controls for clinical genomics

To streamline the article, we have also decided to adopt the term "sequin" (sequencing spike-in), which we use as shorthand for the more cumbersome "chiral DNA standard/control".

We fully agree with the reviewer's aspiration for the development of universal reference standards that fulfil the needs of the genomics community. Therefore, the chiral DNA spike-in controls for human genome sequencing that we describe in the present manuscript, as well as additional sets of sequins for RNA and metagenome sequencing, are freely available for not-for-profit research. Sequins are distributed via a purpose-built website (www.sequin.xyz), and purpose-built software for sequin analysis is also freely available (<https://github.com/student-t/Anaquin>). Indeed, sequins are already distributed to >120 institutes in >20 countries, with this user-community growing rapidly.

In this manuscript we have focused on the conceptual basis, technical development and evaluation of sequins, and we believe it is beyond the scope to discuss their availability, sourcing and distribution in detail (although the establishment of principles and guidelines governing the broad and sustainable distribution of sequins is in progress).

1.2 The paper presents these controls as a scientifically interesting resource for the readers. I note that the website sequin.xyz, presented as a data resource by the authors in this paper, offers aliquots of the controls for use in non-profit research. I suggest that the authors make the availability clear to the reader in the body of the paper.

I offer my strongest recommendation that the authors make the controls universally available, without restriction. Restricted use is inconsistent with principles of transparent and reproducible research, and antithetical with the concept of a "standard."

As just mentioned (see **1.1**), sequins are freely available for not-for-profit research and it is our desire that they become established as a universal standard for genomics. At the reviewer's suggestion, we have added the following statement to the manuscript body:

"By providing commutable internal controls, sequins promise to improve the accuracy and robustness of clinical genome sequencing and, thereby, support the emergence of precision medicine. We offer sequins as a validated resource for the genomics community. For further information, or to request an aliquot please visit www.sequin.xyz."

1.3 Suggestions for additional references to relevant work:

The below noted papers describe prior work in using "spike-in" controls for performance assessment of NGS variant detection -- these would be useful additional references:

Sims, D. J., Harrington, R. D., Polley, E. C., Forbes, T. D., Mehaffey, M. G., McGregor, P. M., ... Lih, C.-J. (2016). Plasmid-Based Materials as Multiplex Quality Controls and Calibrators for Clinical Next-Generation Sequencing Assays. *The Journal of Molecular Diagnostics*, 18(3), 336–349. <https://doi.org/10.1016/j.jmoldx.2015.11.008>

Lincoln, S. E., Zook, J. M., Chowdhury, S., Mahamdallie, S., Fellowes, A., Klee, E. W., ... Lincoln, S. E. (2017). An interlaboratory study of complex variant detection. *BioRxiv*, 301–312. <https://doi.org/https://doi.org/10.1101/218529>

We have incorporated both suggested references into our revised manuscript.

1.4 Minor corrections:

- Page 3: "reaction that amplifying"
- Page 6: "depreciating" -- perhaps "deprecating" or "downsampling"
- Page 7: "microsatellites loci" -- should be "microsatellite loci"
- Page 8: "technically identical" -- suggest "repeated"
- Page 8: "measure and mitigate" -- suggest "assess and mitigate"
- Page 8: "chiral sequences are both commutable..." -- perhaps "chiral sequences are demonstrated here to be commutable..."
- Page 10: State method for concentration determination when pooling controls (UV spectrophotometry?)
- Page 13: "MSI reference samples from NSBCI..." should be "MSI reference samples from NIBSC..."

We have corrected the issues just listed in our revised manuscript.

REVIEWER 2

2.1 The manuscript "Chiral DNA sequences as commutable reference standards for clinical genomics" by Deveson et al. is well structured and written. The authors introduce a very interesting approach using chiral DNA as a standard for next generation sequencing with an emphasis on clinical sequencing. They nicely show that chiral DNA closely shares all tested properties with the corresponding "mirrored original". Nevertheless, when looking closely, one can see some differences between the performance of the original sequence and its chiral twin. These differences are throughout the study often not adequately mentioned and rarely discussed.

Reviewer 2 is correct that coverage distributions, error frequency profiles and variant detection performances are not identical between mirrored human/chiral DNA sequences. However, given the inherent noise in NGS data, one should not expect to obtain identical profiles when performing this comparison. For this reason, in each experiment presented, we also compared performance between technical replicates of identical human DNA samples. The discrepancies observed between replicates indicates the inherent variation in NGS experiments, and sets an upper limit on the similarity that one could reasonably expect to see between mirrored sequences.

Throughout the manuscript we show the properties of matched chiral sequences to be roughly as similar (and sometimes more similar) than between identical human sequences in technical replicates. Therefore, the observed performances of matched chiral sequences are as similar as could be expected, given the inherent variation in NGS assays.

2.2 One of the biggest challenges for clinical routine sequencing is the limited availability of fresh frozen material. Often, samples are directly fixed in formalin and embedded in paraffin (FFPE) for routine assays. This procedure leads to artefacts in NGS sequencing. It would be nice to read what the authors think of this and if there is a possibility to mimic this effect with chiral DNA as well.

We think it is possible to mimic the effects of FFPE fixation in chiral DNA standards by employing similar preparation techniques during manufacture (eg treating synthetic DNA with formalin), however, we have not yet attempted this. One difficulty this approach will face is the fact that the degree of fixation (and severity of the impact on DNA) varies between preparations, making it difficult to match standards to all potential user samples.

Nevertheless, we have used our existing (non-FFPE) standards in conjunction with FFPE-treated samples (and other degraded samples), where they do confer several advantages. Without an internal control, it is difficult to assess the impact of FFPE treatment on a given clinical sample, because artefacts associated with FFPE treatment (which vary between samples) cannot be easily distinguished from artefacts introduced during library preparation, sequencing and analysis (which vary between experiments). By measuring the extent of specific artefacts on non-FFPE chiral DNA sequences within his/her assay, the user can deduce the quality of the starting sample by comparison. Hence, chiral standards allow the user to distinguish errors and biases that derive from the NGS workflow from artefacts that result from sample-related issues (including the impact of FFPE treatment on clinical samples).

At the reviewer's request, we have incorporated a brief discussion of this subject into the revised manuscript:

"Sequins allow users to rapidly assess operational performance and gauge the quality of the accompanying sample. Low-quality samples, such as clinical specimens that are excessively degraded by formalin fixation, can be easily identified by comparison to internal controls, and technical artefacts can be disentangled from meaningful signal, such as mutation signatures in human tumors."

2.3 It has been shown that mutational signatures as well as sequencing errors are context dependent (Alexandrov et al. Cell Rep 2013; Allhoff et al. BMC Bioinformatics 2013), in chiral DNA this context is reverted, I would expect this to be reflected on some properties and would like to see it, at least, discussed in this manuscript.

It is known that the frequency of certain somatic mutations and sequencing errors are elevated at specific short sequence motifs that occur throughout the genome. This is important for the interpretation of cancer genomes, where specific mutational processes (eg UV exposure) leave non-random mutational signatures (ie enrichments of specific base substitutions at specific motifs). Sequencing errors also conform to non-random signatures that must be distinguished from mutation signatures. In this context, chiral DNA reference standards are highly useful, since the internal reference DNA is affected by confounding sequencing errors but not by natural mutation processes, allowing these signatures to be disentangled.

At the reviewer's request, we have incorporated a brief discussion of this subject into the revised manuscript (see **2.2**).

2.4 I could not find the used sequences on <https://sequin.xyz> (only RNA files in the download section, this manuscript is exclusively dealing with DNA).

We apologise for this: the sequin website is currently under maintenance – to be completed this month – and all relevant information/files will be available shortly.

2.5 R2-Differences in Figure 2 are not negligible. The general coverage profiles seem to be comparable but local differences remain. Do these have distinct properties?

We accept the reviewer's assertion that the difference between the R^2 values shown in **Figure 2b** is not negligible. This suggests that the similarity of *fwd/rev* coverage profiles ($R^2 = 0.84$), though very well matched, did not quite equal the concordance observed for identical *fwd/fwd* profiles ($R^2 = 0.94$). We have been unable to identify any specific sequence feature or other systematic explanation for this subtle difference. We accept that our language may have been too strong in this section and, at the reviewer's suggestion, we have made some amendments to more fairly present this comparison. For example:

"...the correlation between per-base coverage profiles for paired *fwd/rev* sequences ($R^2 = 0.84$) was almost as strong as the correlation between identical *fwd/fwd* sequences analyzed in replicate experiments ($R^2 = 0.94$)..."

"...with the concordance of sequencing coverage and error profiles between chiral pairs approaching that of identical human sequences analyzed in technical replicates."

2.6 Did the library of synthetic regions (for hybridization tests) contain SNP-Positions? Did they also synthesize the minor alleles?

The synthetic sequences used for the hybridization analyses presented in **Figure 3** did not contain genetic variants. Therefore, because we compared them to natural human DNA (from NA12878), the human/chiral sequences in this experiment differed at sites of genetic variation in the NA12878 genome (ie they are not perfect chiral counterparts). Given that there are 529 SNVs/indels (as per Genome In A Bottle high confidence annotation) within this 279 kb region (>99% identity), this represents a relatively minor caveat. Nevertheless, we have acknowledged this in the text:

"We then used this custom panel to perform target-enriched NGS on a combined sample containing human genomic DNA (NA12878) and the mixture of synthetic chiral DNA sequences, which differ only at sites of genetic variation in the NA12878 genome (>99% identity; see **Methods**)."

2.7 Did they try aligning the chiral counterpart of the exact same library rather than simulating two independent ones? Does the R2 increase?

To address the reviewer's suggestion, we used the NGS libraries derived from the 8 x 1.8kb synthetic chiral DNA sequence pairs (analysed in **Figure 2**). For each library we retrieved all human reads, and simulated an exact chiral counterpart for each read (including identical Phred quality scores and read-pairing relationships). We then aligned these simulated libraries to the constituent chiral (*fwd/rev*) DNA sequences, and repeated the analyses of coverage similarity that are presented in **Figure 2**.

As the reviewer suggests, the coverage concordance of a human library and a perfectly mirrored simulated library is stronger ($R^2=0.997$; see **Response Fig. 1**, below) than the concordance of two experimental replicate sequencing libraries ($R^2=0.94$). This analysis demonstrates the equivalent performance of sequence alignment between mirrored DNA libraries, in the absence of any technical variation associated with the library preparation and sequencing.

We thank the reviewer for this interesting suggestion and have incorporated this analysis into **Supplementary Fig. 3**.

Response Fig. 1. Matched performance of simulated mirrored sequencing libraries. (a) Normalized sequencing coverage within a single synthetic chiral DNA sequence pairs (1.8 kb), where each human read is matched by a simulated chiral equivalent. (b) Density scatter plot shows the concordance of per-base coverage profiles between simulated mirrored sequencing libraries.

2.8 Where do the NGS-reads from the human sample, that map wrongly to the chiral genome actually map on the reference genome when aligned against the forward genome only? Are there patterns? 0.167% is not negligible especially if this is non-random.

To test this question more thoroughly, we obtained four replicate NA12878 libraries sequenced by Genome in a Bottle project and aligned these to the *hg38-fwd/rev* reference index. Similarly to the result presented in the original manuscript, in two of these samples a small but not negligible number of reads were cross-aligned to *hg38-rev* (0.18% and 0.20%, respectively). However, in the other two samples there were almost no cross-aligned reads ($3.64 \times 10^{-4}\%$, $3.47 \times 10^{-4}\%$). Subsequently, we confirmed (by BLAST search) that the majority of cross-aligned reads in the first two libraries (>99%) were of bacterial origin, and probably represent low-level contamination in these samples. This explains the discrepancy between the four NA12878 libraries, and also the lack of cross-alignment observed for simulated libraries (**Supplementary Table 2**). We have added these new data to **Supplementary Table 2** and clarified this result in the manuscript text:

“Whole-genome NGS libraries from a human sample (NA12878) also exhibited negligible rates of cross-alignment to *hg38-rev*, with the majority of cross-aligned reads originating from low-level bacterial contamination in the samples analyzed (up to 0.20%; **Supplementary Table 2**).”

2.9 The variants in Figure 6b are barely visible. It might be better to show stacked bar plots to show the proportion.

We have edited **Figure 6** to increase the visibility of the variants in panel **b**.

2.10 The idea of a dilution row for different VAFs is very good but unfortunately the sequencing context is not fixed since different VAFs are represented in different genes (at least in Figure 6).

This is correct: it is not possible to represent the same mutation a multiple allele frequencies within a single chiral standard mixture. The VAF ladder instead comprises 94 unique cancer mutations, selected for their clinical relevance, with 7-9 mutations at each VAF level (ranging on a log₂ scale from 100%-0.1%). Whilst individually distinct, the group of variants do replicate many of the context dependent features of mutations of interest in cancer (occur within coding exons etc.).

2.11 It would be nice to see how the chiral standard performs at sites of Indel’s and especially CNVs for cancer genomes, especially deletions that are very hard to determine in target enriched approaches.

While we do not have the relevant data to illustrate the use of chiral DNA standards at the sites of CNVs in cancer genomes, we can provide some examples for indels. Specifically, the cancer genome reference materials (Horizon) that we used to validate our chiral somatic mutation ladder (**Figure 6c**) contain known indels within several cancer-related genes (MET, FLT3, BRCA2 and EGFR). Although we do not have chiral standards that directly represent these specific mutations, standards that cover the relevant regions of these genes provide un-mutated sequence background against which to interpret these challenging variants (see **Response Fig. 2**, below).

In addition, because many of the synthetic chiral variants that we have created were based on known variants in the NA12878 genome, we can provide many examples of matched human/chiral indels from NA12878 samples sequenced with internal chiral standards. These examples show strong similarity between true human indels and synthetic chiral counterparts (see **Response Fig. 3**, below).

Response Fig. 2. Chiral standards provide unmutated background sequence at sites of two known indels (BRCA2:A1689fs, EGFR:ΔE746-A750) in cancer genome reference samples (Horizon).

Response Fig. 3. Examples of matched human/chiral indels from NA12878 genomic DNA sequenced with internal chiral standards.

2.12 It would be nice to get an overview of the actual read counts of all variants (TP as well as FP for both sample and chiral) from the real cancer samples?

The requested read counts are now supplied in **Supplementary Table 4**.

REVIEWER 3

3.1 Chiral DNA as opposed to scrambled DNA or synthetic DNA: The authors indicate that a chiral DNA molecule retains critical context information. This is a reasonable statement but it raises the question of how important context actually is. Certainly, composition has an effect on sequencing performance (GC content for example) but composition can be recapitulated via synthetic or scrambled sequences. Is there any evidence that synthetic or scrambled sequencing have notable different performance? This is a fairly minor question but worth including a few sentences to address.

Nucleotide composition (eg GC-content) and sequence context both have an important impact on the performance on NGS assays. While scrambled synthetic sequences would have the same nucleotide composition as natural counterpart sequences, shuffling abolishes important contextual attributes.

The most obvious example relates to sequence complexity, or repetitiveness. The sequences 5'-AAAACCCCTTTGGGG and 5'-GCGCACTGTACATGTA have the same nucleotide composition but differ greatly in their sequence entropy/complexity. As we have shown in the present manuscript, low-complexity sequences such as microsatellite repeats interfere with the library preparation and sequencing processes, causing low coverage in these regions (**Figure 2a** and **Supplementary Fig. 3c**), are enriched for errors introduced by PCR-amplification (**Figure 2b** and **Figure 6g**) and are refractory to bioinformatic alignment of sequencing reads to the reference genome (**Supplementary Fig. 9b**). Low-complexity sequences also pose problems for hybrid capture during targeted NGS (**Figure 3b** and **Supplementary Fig. 6b**) and accurate base-calling during nanopore sequencing (**Figure 4a,b**). Therefore, a variant located within the first sequence will be more difficult to reliably characterise than a variant located within the second scrambled sequence. By contrast, the mirrored sequence 5'-GGGGTTTCCCAAAA has the same sequence complexity as the first and, as a result, a variant within this sequence will have similar analytic properties. Indeed, for each of the artefacts just mentioned, we have shown that human DNA sequences and synthetic mirrored equivalents are similarly impacted, thereby demonstrating the relevance of sequence context.

While these data was presented in the original manuscript, we have attempted to more explicitly highlight the importance of sequence context in our revised manuscript. For example:

“Synthetic chiral DNA standards can be created to represent almost any human DNA sequence, including analytically-challenging genetic features (such as microsatellite repeats) that are otherwise difficult to reliably assess. By preserving sequence context and nucleotide composition, chiral standards provide faithful analytic proxies for such features.”

3.2 Cost. This is the more important of the questions the authors should address. The authors show improved specificity for true variants at different frequencies. This is a good thing. But it seems that including controls for each and every target (as seems to be what is suggested for capture experiments) would take a large portion of the sequenced bases. This will increase the cost of sequencing to a particular target coverage. Variants are currently validated via low cost Sanger methods. How different are the costs of large scale validations vs taking up sequencing real estate with control sequences.

For the targeted-sequencing experiments presented in **Figure 6**, we used a custom panel targeting 134 cancer-related genes (designed to represent a typical oncology sequencing panel). For every gene with one or more corresponding chiral standards, these were also targeted in our custom panel. In most cases chiral standards cover a handful of exons, not the whole gene, and chiral targets collectively constitute <10% (41.4 kb) of the total panel design (425.8 kb). Accordingly, the benefits presented in **Figure 6** were obtained at a relative increase of ~10% in sequencing cost. Since sequencing costs represent just a fraction of the total cost of the assay, the overall relative increase in cost associated with using chiral standards is even smaller (<5% in this example). While this is not negligible, we believe that the benefits to performance and reproducibility justify this cost, especially given the difficulty of validating somatic variant candidates detected at low variant allele frequencies by Sanger sequencing.

At the reviewer's suggestion, we have incorporated the following discussion of the coverage costs incurred when using spike-in controls:

“The benefits of using chiral DNA standards, or any other spike-in control, during NGS experiments must be weighed against the cost of sequencing reads that are necessarily sacrificed for their analysis. This limitation is of minimal importance during whole-genome sequencing, where no more than ~1% of reads must be sacrificed in order to achieve matched stoichiometry between chiral standards and the diploid human genome. During targeted sequencing approaches, a larger fraction of the sequenced library may be sacrificed, with this depending on the collective size of captured genome regions relative to captured reference sequences, and therefore differing for alternative capture panel designs. For a typical design, users can attain the benefits to performance and reproducibility described above at a 5-10% increase in sequencing cost.”

3.3 I see little use in the section on nanopore. The main thrust of the use of these controls seems to be for clinical applications. Nanopore must make massive changes before this technology could be used for most clinical applications.

Nanopore sequencing currently suffers from a high error rate that makes it non-competitive with Illumina sequencing for the identification of small genetic variants. However, long read-lengths, real-time data acquisition and portability of nanopore devices ensure this emerging technology has significant potential for clinical application.

Further development of nanopore sequencing technology and bioinformatics software for the analysis of nanopore data is required. In this context, the development of an appropriate reference standard (ie chiral DNA standards) for nanopore sequencing could prove highly valuable to the community. Chiral standards represent a rich and well-characterised resource for benchmarking, allowing users to quickly assess the performance of new sequencing devices, reagents, protocols and bioinformatics techniques. Their availability will facilitate the improvement and proliferation of nanopore sequencing (and other emerging sequencing techniques) whilst ensuring robustness and reproducibility. Therefore we disagree that this section of the manuscript, that shows the validity of chiral standards in nanopore sequencing, is without value.

REVIEWERS' COMMENTS:

Reviewer #1 (Remarks to the Author):

The authors have addressed the most significant revisions identified in my review.

I remain disappointed in the authors' lack of commitment to making the control materials available without restriction. I applaud the spirit of making the materials available at no cost for non-profit research, but I expect the restriction from commercial use will significantly diminish the impact of the fine technical work. It's hard to get controls adopted, and commercial application is often the key to overcoming barriers to adoption.

The work is ready for publication as is.

Reviewer #2 (Remarks to the Author):

First of all I would like to congratulate and thank the authors for this interesting manuscript. In my opinion the quality and clarity improved substantially during the revision. All concerns have been addressed accordingly whenever feasible. I do acknowledge that formalin treatment as well as other entirely new experimental approaches are out of the scope of a revision.

There is only very minor remark left.

i) I still cannot find the relevant sequences on the webpage but I am sure this is only a matter of days to be resolved now.

Reviewer #3 (Remarks to the Author):

The author's have addressed my comments adequately. In fact reviewer 2 brought up some very valid critiques that I did not address in my own review. The authors addressed those comments very well and the manuscript is improved for it.

RESPONSE TO REVIEWER COMMENTS

Reviewer 1

The authors have addressed the most significant revisions identified in my review. I remain disappointed in the authors' lack of commitment to making the control materials available without restriction. I applaud the spirit of making the materials available at no cost for non-profit research, but I expect the restriction from commercial use will significantly diminish the impact of the fine technical work. It's hard to get controls adopted, and commercial application is often the key to overcoming barriers to adoption. The work is ready for publication as is.

We thank the reviewer for his/her careful evaluation of our work and helpful feedback throughout. We appreciate the reviewer's view on access for commercial use but we cannot revise our policy at this time.

Reviewer 2

First of all I would like to congratulate and thank the authors for this interesting manuscript. In my opinion the quality and clarity improved substantially during the revision. All concerns have been addressed accordingly whenever feasible. I do acknowledge that formalin treatment as well as other entirely new experimental approaches are out of the scope of a revision.

There is only very minor remark left.

i) I still cannot find the relevant sequences on the webpage but I am sure this is only a matter of days to be resolved now.

We thank the reviewer for his/her careful evaluation of our work and helpful feedback throughout. Our new website (www.sequinstandards.com) has now launched – we apologise for the delay.

Reviewer 3

The author's have addressed my comments adequately. In fact reviewer 2 brought up some very valid critiques that I did not address in my own review. The authors addressed those comments very well and the manuscript is improved for it.

We thank the reviewer for his/her careful evaluation of our work and helpful feedback throughout.